# Genome Wide Identification and Annotation of NGATHA Transcription Factor Family in Crop Plants

**DOI:** 10.3390/ijms23137063

**Published:** 2022-06-25

**Authors:** Hymavathi Salava, Sravankumar Thula, Adrià Sans Sánchez, Tomasz Nodzyński, Fatemeh Maghuly

**Affiliations:** 1Plant Functional Genomics, Institute of Molecular Biotechnology, Department of Biotechnology, BOKU-VIBT, University of Natural Resources and Life Sciences, 1190 Vienna, Austria; hymavathi.salava@gmail.com; 2Mendel Centre for Plant Genomics and Proteomics, Central European Institute of Technology (CEITEC), Masaryk University, Kamenice 5, 625 00 Brno, Czech Republic; 465939@muni.cz (S.T.); adria.sanchez@ceitec.muni.cz (A.S.S.); tomasz.nodzynski@ceitec.muni.cz (T.N.); 3National Centre for Biomolecular Research, Faculty of Science, Masaryk University, Kamenice 5, 625 00 Brno, Czech Republic

**Keywords:** transcription factor, NGATHA (NGA), phylogenetic analysis, evolution, plant development

## Abstract

The NGATHA (NGA) transcription factor (TF) belongs to the ABI3/VP1 (RAV) transcriptional subfamily, a subgroup of the B3 superfamily, which is relatively well-studied in Arabidopsis. However, limited data are available on the contributions of NGA TF in other plant species. In this study, 207 NGA gene family members were identified from a genome-wide search against *Arabidopsis thaliana* in the genome data of 18 dicots and seven monocots. The phylogenetic and sequence alignment analyses divided NGA genes into different clusters and revealed that the numbers of genes varied depending on the species. The phylogeny was followed by the characterization of the Solanaceae (tomato, potato, capsicum, tobacco) and Poaceae (*Brachypodium distachyon*, *Oryza sativa* L. japonica, and *Sorghum bicolor*) family members in comparison with *A. thaliana*. The gene and protein structures revealed a similar pattern for NGA and NGA-like sequences, suggesting that both are conserved during evolution. Promoter *cis*-element analysis showed that phytohormones such as abscisic acid, auxin, and gibberellins play a crucial role in regulating the NGA gene family. Gene ontology analysis revealed that the NGA gene family participates in diverse biological processes such as flower development, leaf morphogenesis, and the regulation of transcription. The gene duplication analysis indicates that most of the genes are evolved due to segmental duplications and have undergone purifying selection pressure. Finally, the gene expression analysis implicated that the NGA genes are abundantly expressed in lateral organs and flowers. This analysis has presented a detailed and comprehensive study of the NGA gene family, providing basic knowledge of the gene, protein structure, function, and evolution. These results will lay the foundation for further understanding of the role of the NGA gene family in various plant developmental processes.

## 1. Introduction

Plant growth and development require numerous rigorous regulatory processes, and therefore, transcriptional regulation plays an important role in every stage of plant growth and development. TFs bind to their target genes or adjacent regions and control gene expression by turning them on and off as needed [1,2], and therefore, they play crucial roles in regulating various plant processes and stress responses. To date, many TF families and their binding sites have been reported [3]. One such family of TFs is the B3 superfamily, which regulates the expression of various genes. B3 proteins are expressed in various plant tissues, suggesting a role for B3 proteins in different plant processes [4]. NGA belongs to the RELATED TO ABI3/VP1 (RAV) transcriptional subfamily, forming a subgroup of the B3 superfamily. The RAV subfamily is further divided into two categories: Class-I (proteins with a B3 domain and an AP2 domain) and Class-II (proteins have a B3 domain but no AP2 domain) [5]. Exceptionally, *Merchantia Polymorpha* has two B3 domains in NGA3 [4]. Although the NGA family has been studied in *A. thaliana* and *Brassica napus*, it has been poorly explored in other plant species.

In Arabidopsis, four *NGA* genes (*AtNGA1*, *AtNGA2*, *AtNGA3*, *AtNGA4*) and three *NGA-LIKE* genes (*AtNGA-LIKE1*, *AtNGA-LIKE2*, *AtNGA-LIKE3*) have been reported [6,7,8,9,10,11,12]. NGA TFs are mainly involved in developing pistils; they are also involved in regulating the shape and size of lateral organs such as leaves and petals and the regulation of seed size [6,8,9,13,14,15,16,17,18]. Alvarez et al. [19] used synthetic miRNAs against *A. thaliana NGA* genes, showing that single *NGA* mutants exhibited mild phenotypic changes in lateral organs, while quadruple mutants exhibited defective pistils as well as small, broad leaves with the broad perianth [19]. Later, in 2009, the same group published a detailed report on the *NGA* genes, *AtNGA1* and *AtNGA4*, expressed in the lateral organs, especially the distal parts [6,20]. The gene *STYLISH1* (*AtSTY1*), known to be involved in carpel development, activates the NGA gene that indirectly regulates gynoecium development. STY1 and NGA co-regulate the *YUCCA2* (*AtYUC2*) and *AtYUC4* genes that promote auxin biosynthesis, thereby regulating the auxin gradient in pistils. In addition, STY1 is known to directly activate *AtYUC4*, suggesting a direct link to auxin biosynthesis [6,20]. This was further confirmed by Trigueros et al. [12]. The same group identified a *tower of pisa-1* (*top1*) mutant by activation markers in Arabidopsis and observed that this mutant was impaired in silique development, with enlarged patterns and reduced fruit size. The *top1* mutant was found to overexpress *AtNGA3* (due to random T-DNA insertions containing the 4x 35S promoter). As a result, the *TOP1*/*NGA3* mutant showed an elongated style [12]. VIGS mediated silencing of *EcNGA* in *Eschscholzia californica* resulted in pistils with impaired style and stigma, but the other parts of the flower such as the petals, sepals, and stamens were unaffected, indicating that NGA is redundant in pistil development. Similarly, *NbNGAa* and *NbNGAb* were downregulated in *Nicotiana benthamiana* using VIGS and it was observed that the length of the style was significantly reduced with improperly fused stigma. It has been documented that the *YUCCA* (*YUC*) genes are involved in auxin biosynthesis and its accumulation in the gynoecium. In Arabidopsis, reduced expression of *YUC* genes was reported in *nga* mutants, resulting in reduced auxin accumulation in the pistils of *nga* mutants [12]. Furthermore, the expression of *NbYUC2* and *NbYUC6* in the apical portion of the gynoecium was decreased in the silenced *NbNGAa* and *NbNGAb* lines, suggesting that the role of *NGA* genes in activating *YUC* genes is involved in promoting auxin gradient across the pistil [7].

NGA TFs also play critical roles in lateral organ growth and development. Lee et al. [8] measured the cell proliferation activity in the lateral organs of overexpression lines and loss-of-function mutants of the NGA family in Arabidopsis, and they observed small, narrow leaves in the overexpression lines and large, wide lateral organs in the quadruple mutants. These results suggest that the NGA family negatively controls cell proliferation in lateral organs [8]. Similar results were observed when *BrNGA1* from Brassica was overexpressed in Arabidopsis, suggesting that *BrNGA1* regulates cell numbers in the lateral organs and roots [21]. This was further supported by the study of Lee et al. [9]. They expressed AtNGA1 in the presence of domains such as CLAVATA3 (CLV3), Meristem layer1 (ML1), WUSCHEL (WUS), SHOOTMERISTEMLESS (STM), and AINTEGUMENTA (ANT) in Arabidopsis. Their results showed that NGA expression in meristems incapacitated pluripotent cells, rendering them incapable of cell differentiation, suggesting an important role for NGA TFs as general differentiation and pistil group identity factors [9].

Despite the limited literature available, NGA has also been involved in various stress responses [10,22]. Sato et al. [10] overexpressed *AtNGA1-GFP* under the influence of the 35S promoter, examined the two-week old seedlings subjected to water stress under the confocal microscope and observed increased protein accumulation of *AtNGA1-GFP* under drought stress compared to the control conditions. They further showed that NGA1 binds to the G-box of the *AtNCED3* promoter under water-deficit conditions to induce ABA biosynthesis. The study also confirmed that the binding of AtNGA1 to the promoter of *AtNCED3* increased under drought stress. A similar pattern was seen in ABA-deficient mutants of Arabidopsis, where NGA induced *AtNCED3* to synthesize ABA in response to stress. In addition, a study by Guo et al. [22] showed that the overexpression of *MtNGA1* from *Medicago truncatula* in *A. thaliana* exhibited increased tolerance to high salt stress. They also exhibited a reduction in the number of branches in the overexpressed lines along with delayed flowering, indicating the importance of NGA as key players in crucial aspects of plant development as well as stress responses. They also examined the reduced shoot branching by analyzing the transcript levels of SMXL genes in the MtNGA1 overexpression lines to observe that the transcript levels of *AtSMXL6*, *AtSMXL7*, and *AtSMXL8* were downregulated while the expression of *AtMAX1/2*, *AtBRC1*, and *AtBRC2* were up-regulated. The repressed shoot branching in the transgenic lines provides important evidence that NGA not only influences ABA, but also regulates strigolactones [22].

To date, phylogenetic analyses of the NGA family of a few plant species such as *A. thaliana*, *B. napus*, *G. max*, *B. distachyon*, *O. sativa*, *P. patens*, and *M. truncatula* have been reported in the literature [10,21]. Furthermore, Pfannebecker et al. [23] combined the phylogeny of members of the NGA family of cruciferous, nightshade, and grass families. Their study concluded that each gene family evolved independently through several rounds of gene duplication events.

In this study, we performed a detailed analysis of the NGA family in higher plant species, focusing on Solanaceae and Poaceae. Phylogenetic reconstruction of the gene family was followed by the characterization of the Solanaceae NGA gene family compared to the monocot members of Poaceae. The characterization included gene and protein structure, protein motifs, promoter analysis, Gene Ontology, and quantitative RT-PCR analysis of the NGA genes. Our obtained data provide a comprehensive understanding of the NGA gene family in higher plants and facilitate further research related to crop plant development and new control methods.

## 2. Results

### 2.1. Identification and Characterization of NGA Genes

We used four NGA and three NGA-Like sequences from *A. thaliana* as the query to identify the NGA sequences in different plant species. An initial search was started with the BLASTP search in phytozome and Ensemble Plants. Databases such as the Sol Genomics Network and Rice Genome Annotation Project were also used to search for NGA family members. Altogether, 460 sequences were retrieved, which were subjected to reciprocal BLASTP against the NGA sequences of *A. thaliana* in the NCBI. The obtained sequences were checked for the presence of the B3 domain (PF02362.21) using the HMM profile of the Pfam and SMART databases. The validated sequences were further assessed using CD-HIT with a threshold of a ≥90% cut-off to eliminate redundant sequences. After the filtering process, 207 sequences of monocots and dicots were obtained to characterize the gene family (Appendix A). The genes were named according to the homology to the Arabidopsis NGA family and the previous literature [6,7,8,9,10,11,12,14,16,18,23].

### 2.2. Phylogenetic Analysis of NGA Family

The evolutionary history of the NGA TF family was investigated by assessing the phylogenetic relationship to classify the NGA proteins. The phylogenetic tree was constructed based on the obtained protein sequences of 18 dicots and 7 monocots (Figure 1; Appendix A). The phylogenetic analysis demonstrated that both the NGA and NGA-Like sequences of dicots and monocots diversified into different clades for (Appendix A). Furthermore, the NGA and NGA-Like sequences of dicots were grouped based on families such as Brassicaceae (*A. thaliana*, *B. rapa*, *Camelina sativa*), Solanaceae (*S. lycopersicum*, *S. tuberosum*, *C. annum*, *N. tabacum*), and other species such as *Populus trichocarpa*, *Phaseolus vulgaris*, *Medicago truncatula*, etc. (Appendix A).

Among the dicots, the number of NGA and NGA-Like protein sequences varied within species, while among the selected monocots, the number of NGA and NGA-Like protein sequences was almost the same; five and two, respectively. However, there were some exceptions: one interesting feature observed here was that *T. aestivum* possessed the highest number of proteins (26), followed by *C. sativa* and *Musa accuminata* (banana) with 20 and 17 sequences, respectively. Furthermore, NGA sequences of banana were grouped separately from other monocots, indicating that banana evolution is independent of other monocots (as showed in Figure 1 and Appendix A). Altogether, our results indicate that the evolution of NGA and NGA-Like sequences have followed divergent lineages. The similarity of protein sequences of *A. thaliana*, *S. lycopersicum*, and *O. sativa* L. japonica was examined to observe that the proteins were 47.63% similar on average (Appendix A).

### 2.3. Physical and Chemical Properties of NGA Family

The physical and chemical properties of the NGA family of Solanaceae (tomato, potato, capsicum, and tobacco) along with Arabidopsis, and Poaceae (rice, sorghum, and Brachypodium) are outlined in Table 1 and Table 2. The table shows the details of the NGA family such as gene ID, chromosome locations, length of the gene, complete coding sequence (CDS), protein molecular weight (MW), and isoelectric point (PI) as well as the predicted location of the signal peptide of the respective proteins. The length of the proteins was between 176 amino acids (CaNGA-Like1-1) to 477 amino acids (CaNGA3), with an average of 324 amino acids. The MW of the proteins ranged from 20.31 kDa (CaNGA-Like1-1) to 52.49 kDa (CaNGA3), with an average of 35.84 kDa (SbNGA1). The predicted IP varied from 4.66 (SbNGA4) to 10.45 (BdNGA5). The IP of twenty proteins was below pH 7, while the IP of twenty-seven proteins was above pH 7. The predicted signal peptides showed that the majority of the NGA proteins were located in the nucleus. In contrast, OsNGA2 and OsNGA4 were located in chloroplast and cytoplasm, respectively (Table 1 and Table 2).

### 2.4. Gene Structure and Protein Motifs Analyses

The gene structure analysis of Solanaceae members followed a similar pattern as that of *A. thaliana* (Figure 2a–c). All the NGA genes possessed single introns with some genes possessing untranslated regions (UTRs) and some without UTRs. Among all of the NGAs, *StNGA3* included the longest 3′ UTR of 4154 bp. Similarly, the *NGA-LIKE* genes contained three exons and two introns with an exception for *AtNGA-LIKE3* and *NtNGA-LIKE* with two exons and one intron as well as *CaNGA-LIKE1-1* with only a single exon and no UTRs. Poaceae members such as rice, sorghum, and Brachypodium also followed a similar fashion as above-mentioned (Figure 2d–f). The *NGA-LIKE* gene in Poaceae with two exons and a single intron was observed in *OsNGA-LIKE1-2*. The number of exons (triple exons and double exons) in a gene is not consistent; however, the results suggest that these genes are conserved in gene structure.

All of the NGA proteins included in the study possessed only one B3 domain (PF02362.21/CL0405). The NGA proteins contained a repressor motif (R/KLFGV) that is responsible for regulating heat stress-related genes (Appendix A) [16,23,24,25,26]. Most of the NGA and NGA-Like proteins possessed the repressor motif except for OsNGA3, cCaNGA-Like1-1, and StNGA-Like 1-1 (Appendix A). These results indicate that NGAs and NGA-Like proteins play an essential role in combating heat stress. We also looked for other protein motifs in the NGAs and NGA-Like sequences using MEME 2.0, represented in Figure 2c,f.

Furthermore, we considered four species of the Solanaceae family (tomato, potato, capsicum, and tobacco), named as “Solanaceae members” in the current study. Three common motifs in the NGA and NGA-Like proteins of the Arabidopsis and Solanaceae members are 1, 2, and 4, representing the conserved B3 domain in these species (Figure 2c and Appendix A). Motifs 7 and 8 are found distributed among the NGA proteins of tomato, potato, capsicum, tobacco, and *A. thaliana*, representing conserved domains specific to NGA proteins, named as the NGA-I and NGA-II domains [7,12]. The repressor motif RLFGV is represented in motif 3, and is involved in various stress responses such as heat, salt, and drought [10,22,24,27]. Other motifs such as motif 16 is unique to NGA3 of *N. tabacum*, while motif 19 is unique to the NGA3 proteins of *C. annum* and *N. tabacum*, indicating the importance of these motifs in plant development processes. Similarly, the unique motifs in CaNGA3, NtNGA3-1, and NtNGA3-2 might take part in various stress responses.

Similarly, motif analysis was also performed with NGA protein sequences in members of Poaceae such as *O. sativa* L. japonica, *B. distachyon*, and *S. bicolor* (Figure 2f and Appendix A), which are named members of Poaceae in this paper. It was observed that three motifs (i.e., motif 1, 2, and 4) are common in both the Solanaceae and Poaceae family members, Arabidopsis and monocots, indicating the conserved B3 domain. Similarly, in Poaceae members, motifs 9 and 10 were conserved only in the NGA sequences, which are named as NGA-II and NGA-I motifs, as described in [7,12]. The repressor motif RLFGV in these monocot species is represented as motif 3 in Figure 2f and Appendix A.

The conserved motifs in the Solanaceae and Poaceae species indicate the perpetuated function of NGA proteins in the plant kingdom. Some conserved motifs such as 11, 12, 14, 16, and 18 are only present in the NGA2 and NGA3 proteins of three monocots, suggesting their conserved role in plant growth and development. Similarly, motifs 6, 7, 8, 13, and 19 are found in the NGA-Like sequences, indicating a conserved function of NGA-Like proteins, possibly in carpel development, and other possible roles. The proteins were also checked for the presence of transmembrane domains using TMHMM-2.0 (https://services.healthtech.dtu.dk/service.php?TMHMM-2.0; accessed on 13 October 2022) and it was predicted that the NGA proteins are not embedded in the transmembranes (Appendix A) [28].

Our results reveal that both Solanaceae and Poaceae members share similar motifs as in Arabidopsis, suggesting a common role of NGAs and NGA-Like sequences. Further experimental analysis would provide more knowledge on the possible roles of these proteins, especially in stress responses.

### 2.5. Promoter cis-Element Analysis of NGA Genes

The promoter *cis*-element analysis was performed as followed by Wei et al. [29]. We considered the 1500 bp upstream region of the NGA genes and looked for the presence of various *cis*-elements that include light, hormones (abscisic acid [ABA], gibberellic acid [GA], auxin, methyl jasmonic acid [MeJA], and salicylic acid [SA]), stress-responsive elements (drought inducibility, defense and stress response), and other *cis*-elements related to anaerobic induction, circadian control, meristem development, flavonoid biosynthesis, low-temperature responsive, zein metabolism, seed-specific regulation, At-rich DNA binding protein, endosperm expression, anoxic specific inducibility, and cell cycle regulation. The details of the promoter *cis*-elements of NGA and NGA-LIKE are represented in Figure 3.

Since the STYLISH1 (STY1) transcription factor is known to bind and induce the NGA gene expression via the DNA binding site ACTCTA(C/A) [31,32,33], we observed the presence of ACTCTAC in the upstream region of the genes, namely, *AtNGA2*, *SlNGA1*, *SbNGA2*, *OsNGA-LIKE1-2*, *SbNGA-LIKE1-1*, *SbNGA-LIKE1-2*, and *BdNGA-LIKE1-2*. At the same time, we identified ACTCTAA in the promoters of AtNGA-LIKE3, SlNGA-LIKE1-3, NtNGA-LIKE1, and OsNGA1. We also checked for other regulatory elements such as CpG islands and tandem repeats using PlantPan (http://plantpan.itps.ncku.edu.tw/; accessed on 22 January 2022) [34]; however, we could not identify either tandem repeats or CpG islands.

### 2.6. Three-Dimensional Structure of NGA Proteins

We analyzed the 3D structure of NGA proteins in Arabidopsis, tomato, and rice (Figure 4 and Appendix A). In Arabidopsis and tomato, we observed that the protein structure had a conserved B3 domain and no AP2 domain, which shows that the NGA structure and function might be conserved in these two species (Appendix A). Conservation of the B3 domain in NGA proteins is a notable feature in the RAV subclade, except for the presence of the AP2 domain, specifically in RAV proteins. Multi-alignment of NGA proteins also confirmed the conserved nature of the B3 domain in Arabidopsis, tomato, and rice, suggesting the functional conservation of these proteins during evolution (Appendix A). Furthermore, another five amino acid motifs (RLFGV, green box in Appendix A) seemed to be present in all of the above three species (i.e., RLFGV), indicating an essential role of this motif in plant development.

### 2.7. Synteny or Gene Duplication Analysis

Even though the NGA family is relatively well-studied in *A. thaliana* compared to other species such as *B. rapa*, the evolution history of the NGA family is not yet understood. In this study, we investigated the evolution and origin of the NGA genes of *S. lycopersicum* in comparison with *A. thaliana* (Figure 5).

We identified five pairs of the syntenic relationship between *A. thaliana* and *S. lycopersicum*, where *AtNGA-LIKE1* and *AtNGA-LIKE2* are both linked to *SlNGA-LIKE1-1* as well as *SlNGA-LIKE1-3*. *AtNGA-LIKE3* is linked to *SlNGA-LIKE1-1*, forming the fifth syntenic pair. Although five genes are paired between *A. thaliana* and *S. lycopersicum*, the number of synteny events suggests the distant evolutionary relationship between these two species (Figure 5a). However, the gene duplication event was also evaluated between *O. sativa* L. japonica and *A. thaliana*, where only one gene pair was observed between *OsNGA-LIKE1-2* and *AtNGA-LIKE2* (Figure 5c). The syntenic relationship between *O. sativa* L. japonica and *A. thaliana* further suggests a distant relationship between these two species. Moreover, despite the distant syntenic relation among the genomes of Arabidopsis, tomato and Arabidopsis, rice, the genes belonging to the same subfamily were linked in each syntenic block, suggesting that these species have evolved from the same ancestor (Figure 5). The syntenic relationship was also assessed in other species such as *B. rapa*, where 32 syntenic pairs were observed, indicating that *B. rapa* is the closest relative of *A. thaliana* (Figure 5b). In addition, gene duplication analysis was also investigated in other dicots such as *P. trichocarpa*, *Vitis vinifera*, *S. tuberosum* and monocots such as *B. distachyon* with 14, 11, 9, 6, and 1 syntenic pairs, respectively (Appendix A). These results show that *P. trichocarpa* is closely related to *A. thaliana* while *B. distachyon* seems to be a distant relative to *A. thaliana* with only one syntenic pair.

We further assessed the association (Ka) and dissociation (Ks) constant of NGA genes to understand the evolutionary rates (Table 3 and Table 4). The Ka/Ks ratio of most of the genes (is less than 1) indicated that the majority of them have evolved slowly under purifying selection pressure. These results indicate that genes have evolved under stringent conditions, thus maintaining the conserved nature of the NGA family during evolution. However, gene pairs of the sorghum NGA gene family, namely, *SbNGA-LIKE1-2* paired with other members of the sorghum NGA family, have resulted from positive Darwinian selection, where the Ka/Ks ratio is greater than 1 (Table 4).

### 2.8. Functional Annotation of the NGA Gene Family

According to the GO analysis, NGA family is predicted to be involved in BPs including leaf shaping (GO:0010358), flower development (GO:0009908), response to karrikin (GO:0080167), meristem maintenance (GO:0010073), seed growth regulation (GO:0080113), glucosinolate metabolic processes (GO:0019760), regulation of leaf morphogenesis (GO:1901371), and the regulation of transcription (GO:0006355) (Figure 6; Table 5). However, MFs include DNA binding (GO:0003677) and protein binding (GO:0005515) while CCs include the nucleus, suggesting that the NGA family transcription factors reside in the nucleus (GO:0005634). The obtained GO data represent that NGA genes play an essential role in regulating lateral organs and the development of gynoecium, and participate in various gene regulations involved in plant development and stress responses.

### 2.9. Expression Analysis of NGA Genes by qPCR

We undertook qPCR to understand the gene expression pattern of the NGA family in Arabidopsis and tomato (Figure 7). *AtNGA1* was highly expressed in cotyledons followed by flowers, where the expression was reduced to half of that in the cotyledons in *A. thaliana*. *AtNGA2* was highly expressed in the rosetta leaf, and the expression was reduced to half in cotyledon, while very minimal expression was observed in the mature leaf. AtNGA3 expression was almost similar in the cotyledon and flower, and there was a gradual decrease in the AtNGA3 expression in mature leaf followed by the rosetta leaf. AtNGA4 shows the highest expression in mature leaf while the expressions were significantly lower in the cotyledon and flower. Similar to *AtNGA4*, *AtNGA-LIKE1*, and *AtNGA-LIKE3* expression was significantly higher in the mature leaf while drastically reduced in the cotyledons, flower, and rosetta leaf (Figure 7).

The *SlNGA* and *SlNGA-LIKE* expression in young tomato leaf was significantly high compared to the cotyledons, flower, and mature leaf (Figure 7). In the cotyledons and mature leaf, *SlNGA* and *SlNGA-LIKE* expression was consistently reduced except for *SlNGA-LIKE1-1* in cotyledon and *SlNGA-LIKE1-3* in mature leaf, where the expression levels were significantly higher. *SlNGA2* showed the highest expression in tomato flower, followed by *SlNGA1* and *SlNGA-LIKE1-3* (Figure 7).

## 3. Discussion

The NGA family belonging to the RAV subfamily of the B3 superfamily is relatively well-characterized in *A. thaliana* compared to other plant species [6,8,9,12,21]. In Arabidopsis, the NGA family is known to be involved in the development of gynoecium and the regulation of lateral organs. However, functional annotation of the NGA family is still an area of limited knowledge. In this study, we performed phylogenetic reconstruction of the NGA family using several dicots (Solanaceae) and monocots (Poaceae) (Figure 1).

The NGA phylogenetic tree has a peculiar feature (i.e., the NGA and NGA-LIKE sequences are very well distinguished, suggesting that these genes have evolved separately with well-demarcated evolution in dicots and monocots (Figure 1)). Furthermore, NGA and NGA-LIKE sequences are defined based on the plant families where members of the Brassicaceae, Solanaceae, and Poaceae are phylogenetically well separated, suggesting that these sequences have resulted from multiple duplication events from the most recent common ancestor. Based on the phylogeny analysis, the NGA sequences from different subfamilies and the number of genes in each species vary. For example, in *B. rapa*, ten NGAs and seven NGA-LIKE genes were present, while in *B. vulgaris*, only one NGA and one NGA-LIKE gene were identified. The highest number of genes were identified in *C. sativa* with 14 NGAs and seven NGA-LIKEs, followed by *T. aestivum* with 18 NGAs and eight NGA-LIKEs (Figure 1; Appendix A). These results indicate that the NGA genes have evolved due to multiple rounds of duplications leading to the expansion of the gene family. Furthermore, among the monocots, banana forms a distinct clade with respect to both the *NGA* and *NGA-LIKE* genes, revealing that the genes within this species might have resulted from repeated segmental duplications (Figure 1 and Appendix A).

Furthermore, the gene structure analysis gives a framework of gene duplications and the functional relationship among the gene families. The exon–intron structures of the NGA family in our analysis revealed that the numbers of exons and introns were conserved among subfamilies, indicating the conserved function of the genes within subfamilies (Figure 2). The same trend has been observed among the protein structures where the NGA and NGA-Like proteins share some common motifs; however, few unique motifs are only present within the subfamilies or unique to species. For example, motifs 9, 11, 13, 14, 15, 17, 18, and 19 were acquired during evolution in the NGA and NGA-Like proteins of the Solanaceae species such as *S. lycopersicum*, *S. tuberosum*, *C. annuum*, and *N. tabacum*, indicating novel functions of the proteins. Similarly, monocots such as *O. sativa* L. japonica, *B. distachyon*, and *S. bicolor* possess common motifs that are also present in Solanaceae members, suggesting a conserved function of the NGA proteins. Consistent with these results, the three-dimensional structure of the proteins was conserved in these species; however, minor alterations in the amino acid sequences contribute to the functional variations among the NGA proteins (Figure 4). The presence of protein motif (RLFGV) in the NGA proteins of *A. thaliana*, *S. lycopersicum*, and *O. sativa* implicates that this motif plays an essential role in plant development (Appendix A). Consistent with this, it has been observed that AtNGA1 possessing the RLFGV motif directly binds to the promoter of *AtNCED3*, thereby inducing ABA biosynthesis in Arabidopsis in response to drought stress (Appendix A) [10]. The presence of the repressor motif is also reported in *N. benthamiana*, *Amborella trichopoda*, and *Aquilegia caerulea* in their respective NGA protein sequences [7]. In addition, this repressor motif is reported to be involved in regulating heat stress in the Heat shock factor B family [7,24,25,27]. These findings indicate the significance of NGA proteins in many aspects of plant development, which is yet to be explored.

The analysis of *cis*-elements in the promoter region of the genes would provide clues into the transcriptional regulation of the respective genes. NGA genes are also looked for in the upstream *cis*-regulatory elements. It has been observed that light-responsive elements are present in the promoters of the genes, suggesting that light plays an important role in regulating these genes (Figure 3). Almost half of the genes were observed to be involved in stress-related responses such as drought inducibility and defense, suggesting that these genes play a role in stress response. The NGA genes also possess hormone response elements such as ABA, GA, MeJA, SA, and auxin. ABA and SA are known to participate in plant stress, and the *cis*-elements analysis indicates that NGA genes might be involved in defense response [36,37,38]. The presence of auxin-responsive elements in the promoters of the NGA genes is an interesting feature. As discussed above, the NGA family regulates the *AtYUC2* and *AtYUC4* genes involved in auxin biosynthesis, especially in carpel development [6,12,16]. However, the direct link of auxin responsive elements with NGA regulation is yet to be discovered. Furthermore, some of these genes also implicate their role in gibberellin signaling and methyl jasmonate pathways. In addition, phytohormone ABA seems to play a major part in carpel development [39,40], and the roles of other hormones such as GA, SA, and MeJA in NGA regulation are still not understood. Among the other *cis*-elements, anaerobic induction and meristem development seem to be majorly involved in the regulating of NGA genes.

Gene duplications are the main source of evolution of gene families, predominantly tandem and segmental duplication events [41]. The synteny analysis of NGA genes of Arabidopsis with tomato, potato, and other species such as *P. trichocapra*, *M. truncatula*, and *O. Sativa* L. japonica showed that most of them have evolved through segmental duplications. However, these duplications are followed by the diversification of gene functions during evolution. In addition, tandem duplications are also not uncommon, as can be seen in the phylogeny with genes or proteins co-existing, resembling their similarities in terms of sequence and functions [42]. The nucleotide variations are the key to evolution within gene families. The Ka/Ks ratio tells us about the synonymous and non-synonymous changes in the gene sequences acquired during evolution and measures the evolutionary pressure of the nucleotide variations within the sequence of the genes [43,44]. The Ka/Ks ratio is assessed in NGA genes. Most of the genes have evolved under negative selection pressure, thereby screening random deleterious mutations, whereas, in *S. bicolor*, each gene pair with *SbNGA-LIKE1-2* showed positive Darwinian selection. Our study revealed that the NGA family has evolved under stringent selection pressure, resulting in the conservation of the gene family.

GO analysis revealed the possible roles of NGA genes in Arabidopsis and tomato. Being derived from the B3 superfamily, NGA is primarily involved in gene regulation by sequence-specific DNA binding activity (including *cis*-elements) and is predicted to be localized in the nucleus (Figure 6; Table 5). As evident from the previous literature on the NGA family in *A. thaliana* and *S. lycopersicum*, the AtNGAs are involved in regulating leaf morphogenesis and flower development [6,9,12,21,45]. In addition, *AtNGA-LIKE1* is thought to be responsive to karrikins, indicating that this gene has a role in Strigolactone signaling. Other functions of *NGA-LIKE* genes include negative regulation of transcription, seed growth regulation, leaf shaping, and meristem maintenance. The possible roles of the NGA gene family based on the Gene Ontology results implicate the potential role of the genes in plant growth, development, and defense. Consistent with the Gene Ontology results, the gene expression of the NGA family in Arabidopsis and tomato reflected the importance of the genes in regulating leaf morphogenesis and flower development (Figure 7). However, the localization of NGA proteins would provide better evidence for protein expression in different cell types rather than gene expression studies. These results correlate with the expression of NGA genes in Arabidopsis and *B. rapa*, affecting the development of lateral organs and floral development [8,9,11,21,46].

## 4. Materials and Methods

### 4.1. Identification of NGA Genes

The amino acid sequences of NGA proteins of *A. thaliana* were obtained from the TAIR website (https://www.arabidopsis.org/download_files/Proteins/TAIR10_protein_lists/TAIR10_pep_20101214; accessed on 19 July 2021). These sequences were used to derive the sequences of tomato (*S. lycopersicum*), potato (*Solanum tuberosum*), capsicum (*Capsicum annum*), and tobacco (*Nicotiana tabacum*) from the Sol Genomics Network (https://solgenomics.net/tools/blast/; accessed on 22 July 2021) using BLASTP (https://blast.ncbi.nlm.nih.gov/Blast.cgi; accessed on 19 July 2021). The sequences in rice (*O. sativa* L. japonica) were retrieved from the Rice Genome Annotation Project (http://rice.uga.edu/analyses_search_blast.shtml; accessed on 24 July 2021). The sequences for other dicots (*Brassica rapa*, *Beta vulgaris*, *Glycine max*, *Helianthus annus*, *Phaseolus vulagaris*, etc.), monocots (*S. bicolor*, *Hordeum vulgare*, *B. distachyon*, *Triticum aestivum*, etc.) and other crop plants were extracted from phytozome (https://phytozome-next.jgi.doe.gov/; accessed on 25 July 2021) and Ensembl Plants (https://plants.ensembl.org/index.html; accessed on 28 July 2021). These sequences were checked for the presence of the B3 domain using the Hidden Markov Model (HMM) profile of the Pfam database (http://pfam.xfam.org/; accessed on 30 July 2021) [47]. The sequences were then subjected to CD-HIT (http://weizhong-lab.ucsd.edu/cd-hit/; accessed on 7 August 2021) and highly similar sequences were filtered from the other low similar sequences. The redundant sequences were filtered from the rest of the sequences. The molecular weights and the isoelectric points were calculated using ProtParam tool-Expasy (https://web.expasy.org/protparam/; accessed on 18 September 2021).

### 4.2. Multiple Sequence Alignment and Phylogenetic Tree Construction

The 207 sequences were aligned by MAFFT using default parameters (https://mafft.cbrc.jp/alignment/server/; accessed on 19 August 2021) [48]. The aligned sequences were used for phylogeny using the neighbor-joining (NJ) method and the bootstrap method with 1000 replicates in MEGA 11 [26].

### 4.3. Exon–Intron Structure and Protein Motif and Structure Analysis

The Gene Structure Display Server 2.0 tool was used to illustrate the location and length of the exons and intron within the respective genes (http://gsds.gao-lab.org/; accessed on 13 September 2021) [49]. The protein motifs were predicted using MEME suite version 5.4.1 (https://meme-suite.org/meme/tools/meme; accessed on 16 September 2021) [50]. The protein length was restricted from 6 to 20 amino acids long, and a maximum of 20 motifs was set.

### 4.4. Three-Dimensional (3D) Structure of NGA Proteins

The 3D structure of the full-length proteins was generated using I-TASSER (https://zhanggroup.org/I-TASSER/; accessed on 18 January 2022). The best threading template close to the target protein was chosen based on the C-score and TM-score [35].

### 4.5. Subcellular Localization of Proteins

The signal peptides of the protein sequences were analyzed by using WoLF PSORT (https://wolfpsort.hgc.jp/; accessed on 20 January 2022) [51] and TargetP-2.0 (https://services.healthtech.dtu.dk/service.php?TargetP-2.0; accessed on 25 January 2022) to estimate the subcellular localization of proteins [52].

### 4.6. Promoter Analysis

The 1500 bp sequence upstream was downloaded for the respective genes. Using PlantCARE (http://bioinformatics.psb.ugent.be/webtools/plantcare/html/; accessed on 30 January 2022) [30], these sequences were examined for the presence of *cis*-elements related to phytohormones, stress, light, and other biological activities. The sequences were also investigated for the presence of CpG islands and tandem repeats using PlantPAN 3.0 (http://plantpan.itps.ncku.edu.tw/; accessed on 22 January 2022) [34].

### 4.7. Chromosomal Location and Gene Duplication and Ontology Analysis

The complete genome, gene, and protein sequences were downloaded from the respective databases for the synteny analysis. The Multiple Collinearity Scan Toolkit (MCScanX) was used to scan the genome to identify the gene duplicated gene pairs [53]. Finally, the orthologous gene pairs were identified using a Dual synteny plotter in TBtools (https://github.com/CJ-Chen/TBtools; accessed on 18 January 2022) [54]. The association and dissociation constants (Ka and Ks) were assessed using the Ka/Ks calculation tool (http://services.cbu.uib.no/tools/kaks; accessed on 22 Fabruary 2022) [55].

### 4.8. Gene Ontology (GO) Analysis

GO enrichment analysis was performed using the Blast2GO tool (https://www.biobam.com/blast2go-basic/; accessed on 25 February 2022) [56] and were categorized into three parts: Molecular Function (MF), Biological Processes (BP), and Cellular Components (CC).

### 4.9. Plant Material

The seeds were surface-sterilized using chlorine gas for four hours and plated on half-strength Murashige and Skoog medium (MS) with 1% sucrose. After 3-day stratification, seedlings were transferred to normal growth condition (150 µmol/m^2^/s, 16/8 h photoperiod, 21 °C and 60% relative humidity). In Arabidopsis, cotyledons, rosetta leaf, mature leaf, and flowers were collected from 7, 21, 28, and 32 day old plants, respectively. In tomatoes, cotyledons, developing young leaves (meristem)m and developed mature leaves (second node from the ground) were collected from 12, 28, and 35 day old plants, respectively. The flowers were obtained from the first set of flowers in both Arabidopsis and tomato.

### 4.10. RNA Extraction and Quantitative (q)PCR Analysis

The RNA from the respective samples was extracted using the Trizol method. First, the contaminating DNA was removed from the extracted RNA using DNase as per the manufacturer’s protocol. The RNA integrity was assessed using the RNA 6000 Pico Kit and Agilent 2100 Bioanalyzer. Finally, one µg of RNA was reverse transcribed to cDNA using the iScript™ cDNA Synthesis Kit (BIO-RAD, CZ). This cDNA was used for qPCR analysis. Quantitative PCR was performed using the SYBR Green PCR Master Mix (Agilent Technologies, Santa Clara, CA, USA) on a 7300 Fast Real-Time PCR system (Applied Biosystems, CA, USA). The primer sequences used for Real-Time PCR were designed using Primer3 software (Appendix A). Ubiquitin and RNaseH were used as the internal controls for Arabidopsis while actin and ubiquitin were used for tomato. The relative expression was calculated using the formula 2(−^ΔΔ^CT), where ^Δ^Ct = (Ct value of target gene) − (Ct value of actin) and ^ΔΔ^CT = ^Δ^Ct of accession −^Δ^Ct of reference.

## 5. Conclusions

The comprehensive analysis of the NGA gene family identified 207 sequences that were classified into different gene families according to species. The identified genes from the selected dicot species (Arabidopsis, tomato, potato, capsicum, and tobacco) and monocot species (rice, sorghum and brachypodium) were characterized for gene structure, protein motif, the 3D structure of proteins, gene duplications, Gene Ontology, and expression studies. The gene structure and protein 3D structure revealed the conserved nature of the gene families across different species. Furthermore, Gene Ontology studies implicated the possible roles of the gene families in various aspects of plant development and stress or defense responses. This is in concordance with the gene expression of the NGA genes, suggesting that the NGA genes are mainly involved in the regulation of lateral organs such as the development of the leaves and flowers. Therefore, the detailed characterization of NGA genes in different species is required for further understanding the gene family in various plant developmental processes.

## Figures and Tables

**Figure 1 ijms-23-07063-f001:**
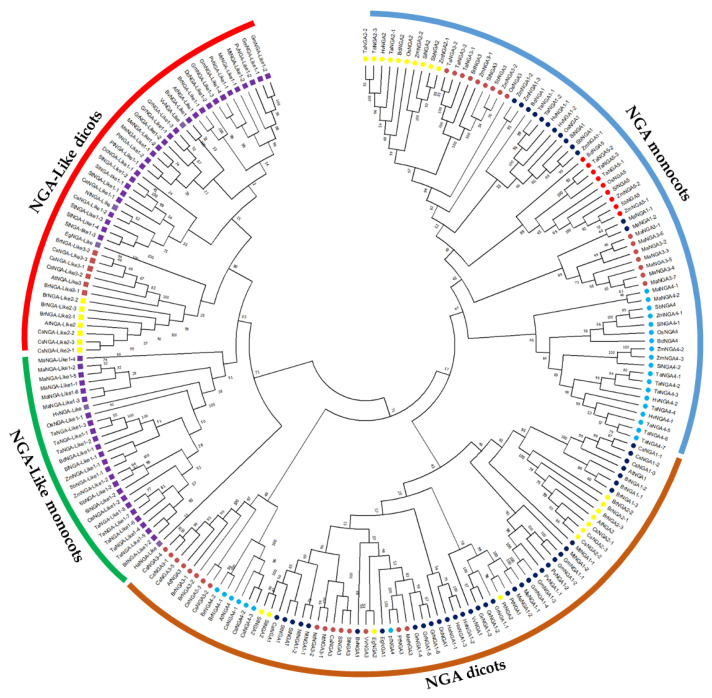
Phylogenetic analysis of the NGATHA family in crop plants. The phylogenetic tree was constructed using the NJ method with 1000 bootstrap replications in MEGA11. The NGATHA proteins were divided into distinct subfamilies with five NGA proteins and three NGA-Like proteins. The symbols are represented to each of the NGATHA proteins as follows. 
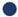
 NGA1, 
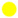
 NGA2, 
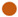
 NGA3, 
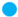
 NGA4, 
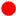
 NGA5, 

 NGA-Like1, 
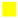
 NGA-Like2, 
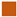
 NGA-Like3.

**Figure 2 ijms-23-07063-f002:**
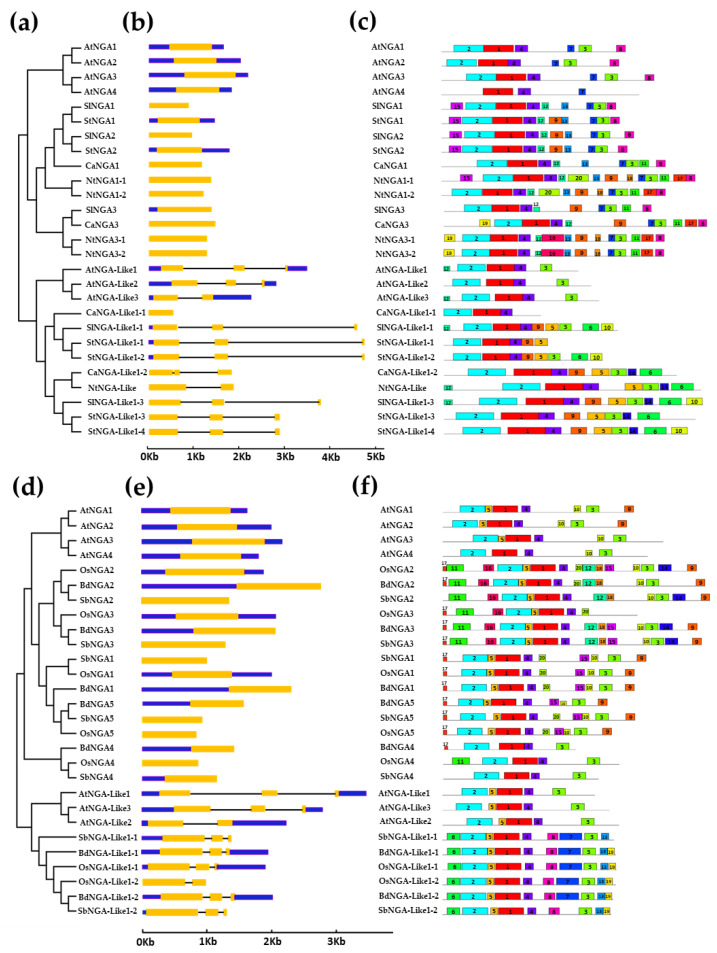
The structural analysis of NGATHA genes and their conserved protein motifs. (**a**,**d**) The phylogenetic tree and classification of NGATHA genes, (**b**,**e**) Exon–intron structures (where exons, introns and untranslated regions are represented by yellow bars, blue bars and black lines, respectively) and (**c**,**f**) protein motifs. (**a**–**c**) represent the gene structures and protein motifs of *A. thaliana*, *S. lycopersicum*, *S. tuberosum*, *C. annuum*, and *N. tabacum*, (**d**–**f**) represent the gene structures and protein motifs of *A. thaliana*, *O. sativa*, *S. bicolor*, and *B. distachyon*.

**Figure 3 ijms-23-07063-f003:**
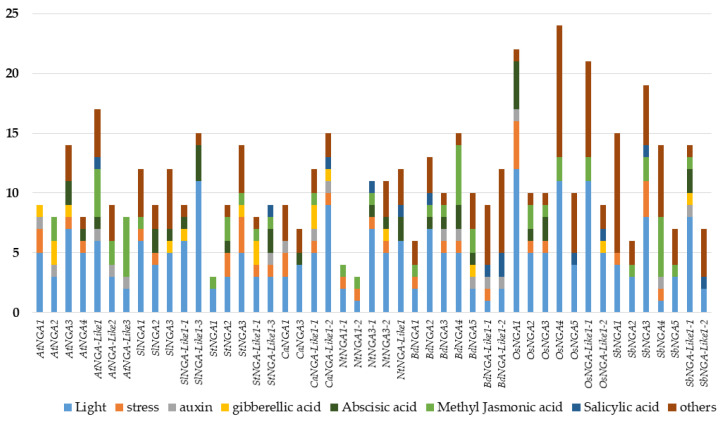
The analysis of *cis*-acting elements in the promoters of NGATHA genes. The X-axis represents the NGATHA genes, and the Y-axis represents the number of *cis*-elements in the promoter of each gene. The *cis*-elements were predicted in the 1500 bp upstream regions using PlantCARE (http://bioinformatics.psb.ugent.be/webtools/plantcare/html/; accessed on 30 January 2022) [30]. “Others” include promoter elements related to anaerobic induction, circadian control, meristem development, flavonoid biosynthesis, zein metabolism, seed specific regulation, endosperm expression and cell cycle regulation.

**Figure 4 ijms-23-07063-f004:**
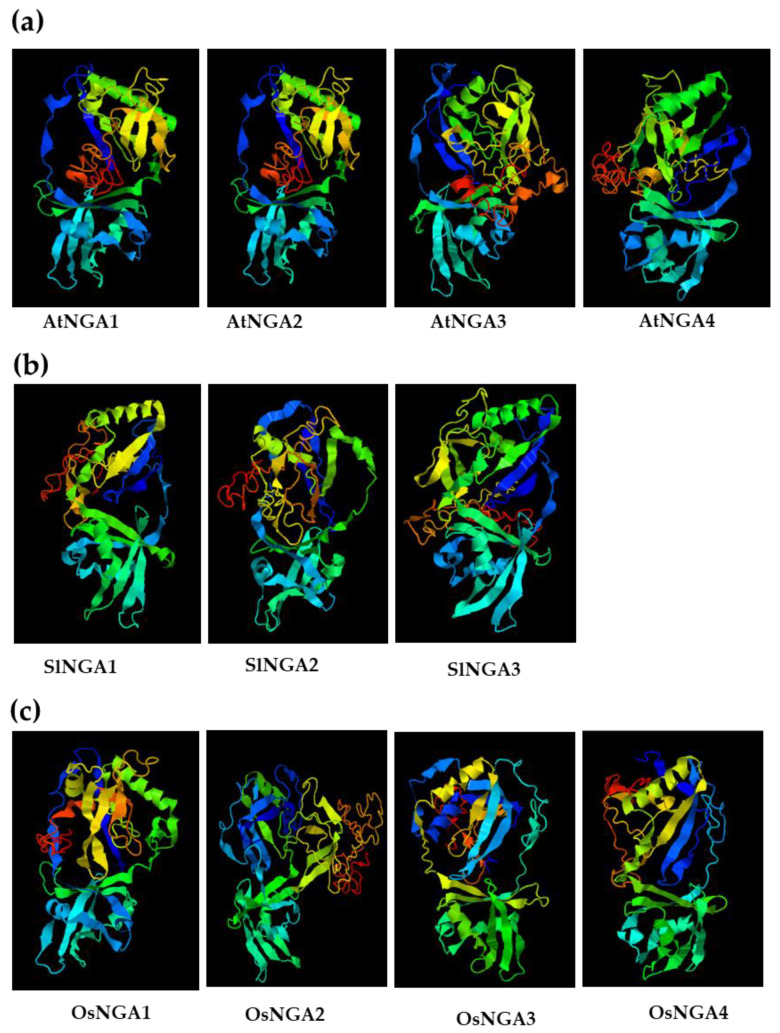
The 3D-structure of the NGATHA proteins in (**a**) *A. thaliana*, (**b**) *S. lycopersicum*, and (**c**) *O. sativa* L. Japonica. The three-dimensional structures were acquired using I-TASSER (https://zhanggroup.org/I-TASSER/; accessed on 18 January 2022) [35].

**Figure 5 ijms-23-07063-f005:**
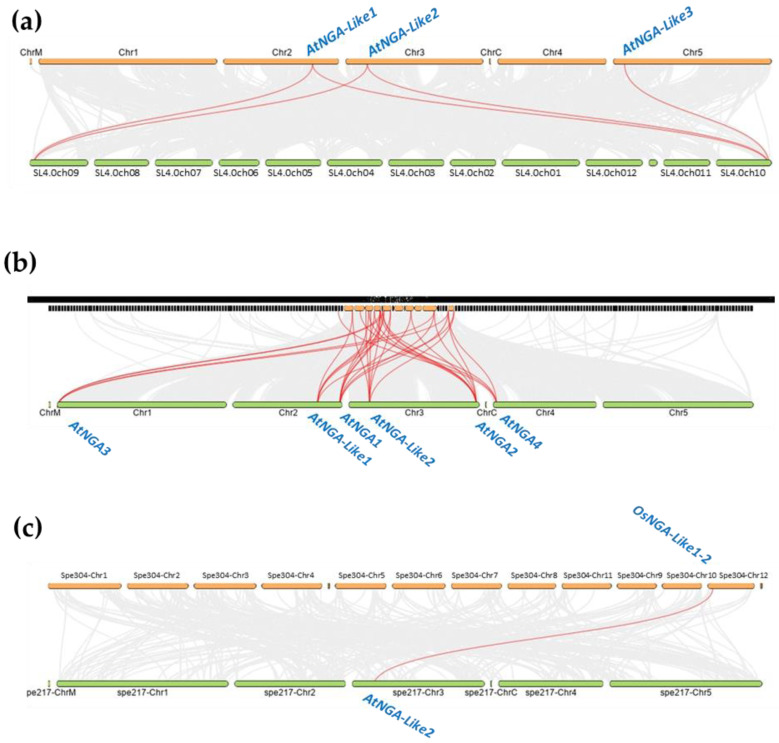
The gene duplication or synteny analysis of the NGATHA genes of *A. thaliana* with (**a**) *S. lycopersicum*; (**b**) *B. rapa,* and (**c**) *O. sativa* L. Japonica. The grey lines (in the background) represent collinear blocks between the respective genomes. The red lines indicate the syntenic gene pairs of *S. lycopersicum*, *B. rapa*, and *O. sativa* L. Japonica with *A. thaliana*.

**Figure 6 ijms-23-07063-f006:**
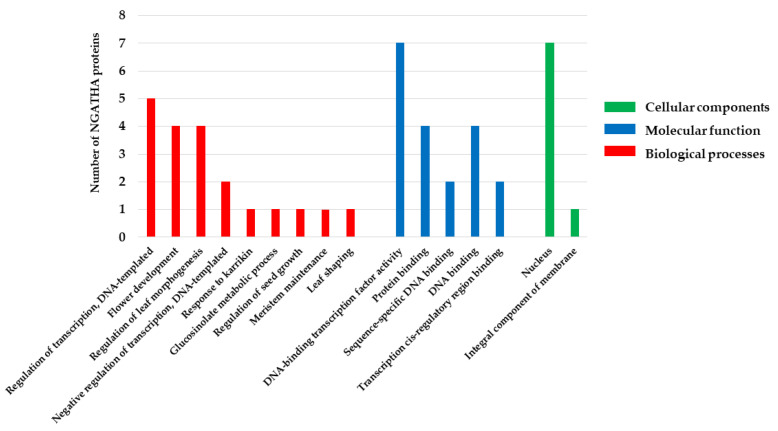
The GO annotation and classification of NGATHA proteins using Blast2GO. Results are summarized for the three main GO categories (BP, MF, and CC). The X-axis includes the most abundant GO terms. The Y-axis represents the number of NGATHA proteins.

**Figure 7 ijms-23-07063-f007:**
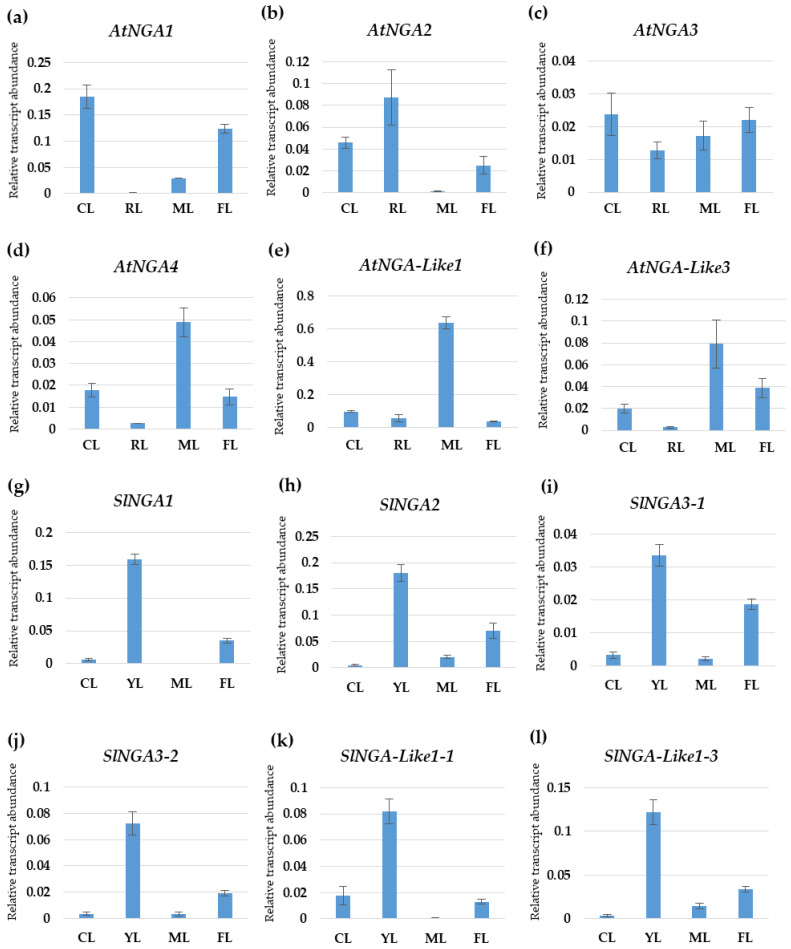
The expression patterns of the NGATHA genes in Arabidopsis (**a**–**f**) and tomato (**g**–**l**) in different plant organs were verified by qPCR. The X-axis represents different plant organs: CL—cotyledon, RL—rosetta leaf, FL—flower, YL—young leaf, ML—mature leaf. The Y-axis represents the relative transcript abundance of the respective genes.

**Table 1 ijms-23-07063-t001:** The list of NGATHA genes in Arabidopsis and Solanaceae (tomato, potato, capsicum, and tobacco) and the features of each gene and protein.

Transcript ID	Gene Name	Chromosome	Chromosome Locus	Strand	Gene Length (bp)	CDS Length (bp)	Protein Length (aa)	Protein Weight (kDa)	pI	Signal Peptide
AT2G46870.1	*AtNGA1*	2	Chr2:19260906..19262533	F	1628	933	310	34.88	6.75	Nucleus
AT3G61970.1	*AtNGA2*	3	Chr3:22951323..22953265	F	1943	900	299	34.27	8.94	Nucleus
AT1G01030.1	*AtNGA3*	1	Chr1:11649..13714	R	2066	1077	358	40.28	6.24	Nucleus
AT4G01500.1	*AtNGA4*	4	Chr4:639247..640976	F	1730	1002	333	38.51	8.52	Nucleus
AT2G36080.1	*AtNGA-LIKE1*	2	Chr2:15148259..15151634	R	3376	735	244	28.43	6.80	Nucleus
AT3G11580.1	*AtNGA-LIKE2*	3	Chr3:3648902..3651726	R	2825	804	267	30.21	7.33	Nucleus
AT5G06250.1	*AtNGA-LIKE3*	5	Chr5:1891880..1894353	R	2162	849	282	31.64	6.97	Nucleus
Solyc08g013700.1.1	*SlNGA1*	8	SL4.0ch08:3090386..3091270	F	885	885	294	33.70	8.36	Nucleus
Solyc08g013690.1.1	*SlNGA2*	8	SL4.0ch08:3081975..3082949	F	975	975	324	36.67	7.72	Nucleus
Solyc05g004000.1.1	*SlNGA3*	5	SL4.0ch05:12418..14381	F	1964	1164	378	42.49	6.50	Nucleus
Solyc10g083210.2.1	*SlNGA-LIKE1-1*	10	SL4.0ch10:62238789..62243955	F	5167	927	316	36.15	6.59	Nucleus
Solyc09g010230.2.1	*SlNGA-LIKE1-3*	9	SL4.0ch09:3659577..3663391	F	3815	1038	345	39.11	7.02	Nucleus
PGSC0003DMP400010383	*StNGA1*	8	ST4.03ch08:5461993..5463392	R	1400	903	300	34.27	9.01	Nucleus
PGSC0003DMP400010384	*StNGA2*	8	ST4.03ch08:5469187..5470896	R	1710	942	313	35.74	7.16	Nucleus
PGSC0003DMP400021619	*StNGA3*	8	ST4.03ch08:56844094..56849780	F	5687	1221	406	45.64	6.27	Nucleus
PGSC0003DMP400048918	*StNGA-LIKE1-1*	10	ST4.03ch10:55899618..55904610	R	4993	867	189	21.28	6.97	Nucleus
PGSC0003DMP400048917	*StNGA-LIKE1-2*	10	ST4.03ch10:55899618..55904610	R	4993	867	288	32.64	6.39	Nucleus
PGSC0003DMP400015721	*StNGA-LIKE1-3*	9	ST4.03ch09:2309350..2314043	R	2774	1008	335	37.22	6.81	Nucleus
PGSC0003DMP400015722	*StNGA-LIKE1-4*	9	ST4.03ch09:2309350..2314043	R	2774	1008	325	36.29	6.52	Nucleus
CA01g34800	*CaNGA1*	1	Pepper1.55ch01:272613133..272614266	F	1134	1134	377	42.48	7.79	Nucleus
CA01g00060	*CaNGA3*	1	Pepper1.55ch01:132987..134420	R	1434	1434	477	52.49	6.16	Nucleus
CA10g17390	*CaNGA-LIKE1-1*	10	Pepper1.55ch10:223101144..223101674	R	531	531	176	20.31	6.12	Nucleus
CA09g00050	*CaNGA-LIKE1-2*	9	Pepper1.55ch09:519882..521660	F	1779	933	310	35.27	5.91	Nucleus
Nitab4.5_0000799g0050.1	*NtNGA1-1*	Nt15	Nitab4.5_0000799:325787..327070	F	1284	1284	427	47.39	8.21	Nucleus
Nitab4.5_0005519g0040.1	*NtNGA1-2*	--	Nitab4.5_0005519:96109..97242	R	1134	1134	377	42.00	7.32	Nucleus
Nitab4.5_0015619g0010.1	*NtNGA3-1*	--	Nitab4.5_0015619:684..1886	R	1203	1203	400	44.91	6.36	Nucleus
Nitab4.5_0012411g0010.1	*NtNGA3-2*	--	Nitab4.5_0012411:19665..20864	R	1200	1200	399	44.71	6.33	Nucleus
Nitab4.5_0000573g0030.1	*NtNGA-LIKE*	Nt10	Nitab4.5_0000573:193812..195562	R	1751	1029	342	38.89	5.96	Nucleus

**Table 2 ijms-23-07063-t002:** The list of NGATHA genes in Poaceae (*O. sativa* L. Japonica, *B. distachyon*, and *S. bicolor*) and the features of each gene and protein.

Transcript ID	Gene Name	Chromosome	Start	End	Strand	Gene Length (bp)	CDS Length (bp)	Protein Length (aa)	Protein Weight (kD)	pI	Signal Peptide
LOC_Os03g02900.1	*OsNGA1*	3	1152918	1154876	F	1959	936	312	33.77	10.28	Nucleus
LOC_Os02g45850.1	*OsNGA2*	2	27934689	27932831	R	1859	1239	413	43.94	8.13	Chloroplast
LOC_Os04g49230.1	*OsNGA3*	4	29368515	29366456	R	2060	951	317	34.50	8.72	Nucleus
LOC_Os08g06120.1	*OsNGA4*	8	3377469	3378332	F	864	864	288	30.71	4.80	Cytoplasm
LOC_Os10g39190.1	*OsNGA5*	10	20917099	20916161	R	939	939	313	33.12	10.03	Nucleus
LOC_Os11g05740.1	*OsNGA-LIKE1-1*	11	2633563	2631707	R	1857	840	280	30.96	7.19	Nucleus
LOC_Os12g06080.1	*OsNGA-LIKE1-2*	12	2836995	2836021	R	1857	834	278	31.07	7.78	Nucleus
SORBI_3001G528200	*SbNGA1*	1	79216533	79218457	R	4767	993	330	35.84	9.83	Nucleus
SORBI_3004G280500	*SbNGA2*	4	62291757	62293326	F	1570	1305	434	45.77	9.08	Nucleus
SORBI_3006G190400	*SbNGA3*	6	54458421	54463486	R	5066	1263	420	44.76	6.31	Nucleus
SORBI_3007G047500	*SbNGA4*	7	4761522	4763508	F	2874	762	253	27.37	4.66	Nucleus
SORBI_3001G313800	*SbNGA5*	1	60096593	60097417	F	825	825	274	29.62	10.44	Nucleus
SORBI_3005G041400	*SbNGA-LIKE1-1*	5	3828689	3830488	R	2277	825	274	30.78	7.20	Nucleus
SORBI_3008G041100	*SbNGA-LIKE1-2*	8	3974205	3975994	R	1790	963	321	35.33	8.64	Nucleus
BRADI_1g77150v3	*BdNGA1*	1	73755394	73760932	R	6207	936	312	33.84	9.87	Nucleus
BRADI_3g51840v3	*BdNGA2*	3	52606532	52612120	R	6309	1281	426	45.39	9.91	Nucleus
BRADI_5g19260v3	*BdNGA3*	5	22424520	22430521	R	6393	1242	413	44.03	6.50	Nucleus
BRADI_3g16500v3	*BdNGA4*	3	14677720	14679110	R	1760	651	216	23.82	5.16	Nucleus
BRADI_3g32140v3	*BdNGA5*	3	34148655	34150224	R	1937	804	267	29.17	10.45	Nucleus
BRADI_4g25170v3	*BdNGA-LIKE1-1*	4	30327514	30329491	F	1978	834	277	31.13	7.34	Nucleus
BRADI_4g42167v3	*BdNGA-LIKE1-2*	4	46194232	46196263	R	2031	1812	273	30.47	7.21	Nucleus

**Table 3 ijms-23-07063-t003:** Synonymous and non-synonymous substitutions in Arabidopsis and Solanaceae members (tomato, potato, and capsicum).

Seq_1	Seq_2	Ka	Ks	Ka/Ks
*AtNGA2*	*AtNGA1*	0.239463	1.069787	0.223842
*AtNGA3*	*AtNGA2*	0.431593	2.061262	0.209383
*AtNGA3*	*AtNGA1*	0.380109	1.735817	0.21898
*AtNGA4*	*AtNGA2*	0.433425	2.596013	0.166958
*AtNGA4*	*AtNGA1*	0.581486	1.806179	0.321943
*AtNGA4*	*AtNGA3*	0.325508	0.955495	0.340669
*AtNGA-LIKE1*	*AtNGA1*	0.525409	2.717022	0.193377
*AtNGA-LIKE1*	*AtNGA3*	0.583534	2.705568	0.215679
*AtNGA-LIKE2*	*AtNGA-LIKE1*	0.312877	1.701328	0.183901
*AtNGA-LIKE2*	*AtNGA4*	0.582629	2.728311	0.213549
*AtNGA-LIKE2*	*AtNGA3*	0.599151	2.270156	0.263925
*AtNGA-LIKE2*	*AtNGA1*	0.616917	2.13734	0.288638
*AtNGA-LIKE3*	*AtNGA-LIKE1*	0.317519	1.9584	0.162132
*AtNGA-LIKE3*	*AtNGA-LIKE2*	0.278815	0.997563	0.279496
*AtNGA-LIKE3*	*AtNGA3*	0.645778	2.283392	0.282815
*AtNGA-LIKE3*	*AtNGA4*	0.782203	2.740771	0.285395
*AtNGA-LIKE3*	*AtNGA1*	0.605938	2.043846	0.29647
*AtNGA-LIKE3*	*AtNGA2*	0.648893	1.939374	0.334589
*SlNGA1*	*SlNGA3-2*	0.377951	1.728417	0.218669
*SlNGA2*	*SlNGA1*	0.106566	0.186894	0.570193
*SlNGA2*	*SlNGA3-2*	0.38381	1.879875	0.204168
*SlNGA3-1*	*SlNGA1*	0.407872	1.656093	0.246286
*SlNGA3-1*	*SlNGA2*	0.401201	1.790513	0.224071
*StNGA2*	*StNGA1*	0.098184	0.216778	0.452926
*StNGA3*	*StNGA2*	0.33704	1.277473	0.263833
*StNGA3*	*StNGA1*	0.321476	1.566063	0.205276
*CaNGA1*	*CaNGA-LIKE1-1*	0.43443	2.62465	0.165519
*CaNGA3*	*CaNGA1*	0.289584	1.43828	0.201341
*CaNGA-LIKE1-2*	*CaNGA-LIKE1-1*	0.288739	3.138935	0.091986

**Table 4 ijms-23-07063-t004:** Synonymous and non-synonymous substitutions in Poaceae members (rice, sorghum, and Brachypodium).

Seq_1	Seq_2	Ka	Ks	Ka/Ks
*OsNGA1*	*OsNGA2*	0.305494	0.628893	0.485764
*OsNGA1*	*OsNGA-LIKE1-2*	0.54703	0.839035	0.651975
*OsNGA1*	*OsNGA4*	0.504384	0.632214	0.797806
*OsNGA2*	*OsNGA-LIKE1-1*	0.564957	1.040672	0.542877
*OsNGA2*	*OsNGA4*	0.453742	0.716875	0.632945
*OsNGA3*	*OsNGA2*	0.26593	0.809453	0.32853
*OsNGA3*	*OsNGA5*	0.420666	0.887946	0.473751
*OsNGA3*	*OsNGA1*	0.332406	0.666317	0.498871
*OsNGA3*	*OsNGA-LIKE1-1*	0.582686	0.811797	0.717773
*OsNGA3*	*OsNGA-LIKE1-2*	0.681818	0.736576	0.925658
*OsNGA4*	*OsNGA3*	0.551308	0.829274	0.664809
*OsNGA4*	*OsNGA-LIKE1-2*	0.609499	0.803316	0.758729
*OsNGA5*	*OsNGA1*	0.257766	0.511455	0.503987
*OsNGA5*	*OsNGA4*	0.475188	0.857441	0.554194
*OsNGA5*	*OsNGA2*	0.389599	0.678508	0.574199
*OsNGA5*	*OsNGA-LIKE1-2*	0.493971	0.671741	0.73536
*OsNGA-LIKE1-1*	*OsNGA-LIKE1-2*	0.124596	0.409362	0.304365
*OsNGA-LIKE1-1*	*OsNGA4*	0.576345	1.095455	0.526124
*OsNGA-LIKE1-1*	*OsNGA1*	0.53016	0.993764	0.533487
*OsNGA-LIKE1-1*	*OsNGA5*	0.605699	0.762845	0.794
*OsNGA-LIKE1-2*	*OsNGA2*	0.516386	0.730013	0.707365
*SbNGA1*	*SbNGA5*	0.273197	0.799784	0.341589
*SbNGA1*	*SbNGA-LIKE1-1*	0.561322	1.413283	0.397176
*SbNGA1*	*SbNGA4*	0.486256	0.900232	0.540145
*SbNGA2*	*SbNGA3*	0.211713	0.648596	0.326418
*SbNGA2*	*SbNGA-LIKE1-1*	0.6945	1.682276	0.412833
*SbNGA2*	*SbNGA1*	0.357362	0.733581	0.487147
*SbNGA2*	*SbNGA4*	0.533568	1.083899	0.492267
*SbNGA2*	*SbNGA-LIKE1-2*	2.151221	0.983169	2.188048
*SbNGA3*	*SbNGA1*	0.346182	0.622191	0.556391
*SbNGA3*	*SbNGA-LIKE1-2*	2.297187	1.00679	2.281695
*SbNGA4*	*SbNGA-LIKE1-1*	0.578696	1.494752	0.387152
*SbNGA4*	*SbNGA5*	0.504508	1.260005	0.400402
*SbNGA4*	*SbNGA3*	0.524943	0.934137	0.561955
*SbNGA5*	*SbNGA2*	0.354755	1.071084	0.331211
*SbNGA5*	*SbNGA3*	0.350202	0.838395	0.417705
*SbNGA-LIKE1-1*	*SbNGA3*	0.597774	1.547476	0.38629
*SbNGA-LIKE1-1*	*SbNGA5*	0.576339	1.47595	0.390487
*SbNGA-LIKE1-2*	*SbNGA-LIKE1-1*	2.471022	1.784932	1.384379
*SbNGA-LIKE1-2*	*SbNGA4*	2.476792	1.359158	1.8223
*SbNGA-LIKE1-2*	*SbNGA5*	2.565304	1.129359	2.271469
*SbNGA-LIKE1-2*	*SbNGA1*	2.799022	0.999304	2.800971
*BdNGA1*	*BdNGA3*	0.346119	0.955767	0.362138
*BdNGA1*	*BdNGA4*	0.514432	1.005848	0.511441
*BdNGA1*	*BdNGA5*	0.377561	0.629852	0.599444
*BdNGA1*	*BdNGA-LIKE1-1*	0.577821	0.928236	0.622493
*BdNGA2*	*BdNGA1*	0.357956	0.705882	0.507104
*BdNGA2*	*BdNGA-LIKE1-1*	0.601821	1.157722	0.519832
*BdNGA3*	*BdNGA2*	0.253051	0.605726	0.417765
*BdNGA4*	*BdNGA-LIKE1-2*	0.570769	1.455996	0.392013
*BdNGA4*	*BdNGA2*	0.437618	1.053323	0.415464
*BdNGA4*	*BdNGA3*	0.461456	0.978035	0.47182
*BdNGA4*	*BdNGA-LIKE1-1*	0.518673	1.080127	0.480196
*BdNGA4*	*BdNGA5*	0.477401	0.865785	0.551408
*BdNGA5*	*BdNGA3*	0.411125	0.937938	0.438328
*BdNGA5*	*BdNGA-LIKE1-2*	0.650776	1.243835	0.523201
*BdNGA5*	*BdNGA2*	0.417842	0.646229	0.646585
*BdNGA-LIKE1-1*	*BdNGA-LIKE1-2*	0.111905	0.656472	0.170464
*BdNGA-LIKE1-1*	*BdNGA3*	0.610294	1.109897	0.549865
*BdNGA-LIKE1-1*	*BdNGA5*	0.641699	0.886495	0.72386
*BdNGA-LIKE1-2*	*BdNGA3*	0.544777	1.634145	0.333371
*BdNGA-LIKE1-2*	*BdNGA1*	0.611541	1.29977	0.470499
*BdNGA-LIKE1-2*	*BdNGA2*	0.593526	1.237082	0.479779

**Table 5 ijms-23-07063-t005:** The GO classification of the annotated NGATHA genes in Arabidopsis and tomato.

	GO Term	Annotation	Involved Genes
BP	GO:0006355	Regulation of transcription, DNA-templated	*AtNGA1*, *AtNGA2*, *AtNGA3*, *AtNGA4*, *AtNGA-LIKE2*
	GO:0009908	Flower development	*AtNGA1*, *AtNGA2*, *AtNGA3*, *AtNGA4*
	GO:1901371	Regulation of leaf morphogenesis	*AtNGA1*, *AtNGA2*, *AtNGA3*, *AtNGA4*
	GO:0045892	Negative regulation of transcription, DNA-templated	*AtNGA-LIKE1*, *AtNGA-LIKE3*
	GO:0080167	Response to karrikin	*AtNGA-LIKE1*
	GO:0019760	Glucosinolate metabolic process	*AtNGA-LIKE2*
	GO:0080113	Regulation of seed growth	*AtNGA-LIKE2*
	GO:0010073	Meristem maintenance	*AtNGA-LIKE3*
	GO:0010358	Leaf shaping	*AtNGA-LIKE3*
MF	GO:0003700	DNA-binding transcription factor activity	*AtNGA1*, *AtNGA2*, *AtNGA3*, *AtNGA4*, *AtNGA-LIKE1*, *AtNGA-LIKE2*, *AtNGA-LIKE3*
	GO:0005515	Protein binding	*AtNGA1*, *AtNGA3*, *AtNGA4*
	GO:0043565	Sequence-specific DNA binding	*AtNGA1*
	GO:0003677	DNA binding	*AtNGA2*, *AtNGA3*, *AtNGA4*, *AtNGA-LIKE3*
	GO:0000976	Transcription *cis*-regulatory region binding	*AtNGA-LIKE1*, *AtNGA-LIKE2*
CC	GO:0005634	Nucleus	*AtNGA1*, *AtNGA2*, *AtNGA3*, *AtNGA4*, *AtNGA-LIKE1*, *AtNGA-LIKE2*, *AtNGA-LIKE3*
	GO:0016021	Integral component of membrane	*AtNGA-LIKE2*

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
