# Peer review of "Genome Wide Identification and Annotation of NGATHA Transcription Factor Family in Crop Plants"

_ijms, 2022, doi:10.3390/ijms23137063_

Round 1

Reviewer 1 Report

I checked your manuscript and described comments below.

This paper does not have figures, tables and supplementary materials.

I think the author should repost including these contents and files

Author Response

Thank you very much for your review. Here are the responses. 

I checked your manuscript and described the comments below.

Thank you!

This paper does not have figures, tables and supplementary materials.

We are very sorry; all figures and tables were uploaded separately in the first submission and they were not in the main text. In the resubmission, we used the journal template and we added all figures and tables in the main text. Supplementary figures and tables are in separate files.

I think the author should repost these contents and files

Corrected as suggested.

Reviewer 2 Report

It looks like inetresting topic,7

however so far I have found only one .pdf file without figures tables, etc.

Please, for the second round provide full text incuding figures,tables, etc.

Short rather formal comments:

Line 24 : please, change punctuation, mention light separately. 

Line 30: it is better to write analysis, not study.

Lines 444-447: please, clarify conditions: soil, medium etc. Which genotypes have been used? Comparison of 10 days old Arabidopsis and tomato does not have too much sense: the have different developmental stages. For example, 10 days old cotyledon already passed several round of endocycle have strong DNA and histone methylation. This is not case for tomato.

Line 452: „The RNA extracted was used for qPCR analysis“- RNA or cDNA???

Line 456: why you have used different reference genes for tomato and Arabidopsis?

Author Response

Thank you very much for the review. Please find the responses below.

It looks like inetresting topic.

Thank you!

however so far I have found only one .pdf file without figures tables, etc.

We are very sorry; all figures and tables were uploaded separately in the first submission and they were not in the main text.

Please, for the second round provide full text including figures, tables, etc.

In the resubmission, we used the journal template and we added all figures and tables in the main text. Supplementary figures and tables are in separate files.

Short rather formal comments:

Line 24: please, change punctuation, mention light separately. 

Modified as suggested.

Line 30: it is better to write analysis, not study.

Modified as suggested.

Lines 444-447: please, clarify conditions: soil, medium etc. Which genotypes have been used? Comparison of 10 days old Arabidopsis and tomato does not have too much sense: the have different developmental stages. For example, 10 days old cotyledon already passed several rounds of endocycle have strong DNA and histone methylation. This is not case for tomato.

Thank you! As requested, we added the necessary information. We also corrected our typo mistake since the samples from Arabidopsis thaliana were collected from the seven-day-old seedlings.  “The seeds were surface-sterilized using chlorine gas for four hours and plated on half-strength Murashige and Skoog medium (MS) with 1% sucrose. After 3-day stratification, plants are transferred to normal growth condition (150 µmol/m2/s, 16/8h photoperiod, 21°C and 60% relative humidity). The seedlings and plants are maintained at a 16/8-hour photoperiod, 21°C and 60% relative humidity. In Arabidopsis, cotyledons, flowers, rosetta and secondary leaves of seven-day-old seedlings are collected. In tomatoes, cotyledons, flowers, young and mature leaves are collected from 10-days old seedlings.”

Line 452: „The RNA extracted was used for qPCR analysis“- RNA or cDNA???

Finally, one µg of RNA is reverse transcribed to cDNA using iScript™ cDNA Synthesis Kit (BIO-RAD, CZ). This cDNA is used for qPCR analysis.

Line 456: why you have used different reference genes for tomato and Arabidopsis?

We used the housekeeping genes (actin, ubiquitin and RNaseH) for our real-time PCR as internal controls.

Reviewer 3 Report

The manuscript is valuable, but with some unfinished discussion part. Rather presents obvious results.

Author Response

Sorry for the late response. Thank you for the valuable suggestions. 

We went through the whole manuscript for clarification, as requested.

Two marked sentences should be composed in one.

Modified as suggested: “: Class-I and Class-II. Class-I proteins include proteins with a B3 domain and an AP2 domain, while class-II proteins have a B3 domain but no AP2 domain [5].”

Source needed.

Added as requested.

A better formulation is needed.

Modified as suggested: “We used four NGA and three NGA-Like sequences from A. thaliana, as the query, to identify the NGA sequences in different plant species.”

It is named Solanaceae by botany classification. No need to explain it.

Corrected as suggested: “Besides, we have considered four species of the Solanaceae family (tomato, potato, capsicum and tobacco), named as “Solanaceae members” in the current study.”

Not needed to explain that conclusion is in brief.

Thank you, as suggested, we removed it.

Round 2

Reviewer 1 Report

I checked your revised manuscript and described comments below.

I have confirmed that this manuscript contains figures, tables and supplementary materials.

I think this paper is a good analysis of the NGA gene familiy. Especially, good point is to conclude that NGA gene familiy is related to plant developmental processes.

I don't think this paper has any major mistakes or grammatical problems.

Author Response

We are grateful to review for careful reading, helpful comments, and constructive suggestions, which has significantly improved our manuscript. 

Reviewer 2 Report

Thank you for the new version.

The text is improved, but some questions still need to be answered. The most problematic point is NGA gene localiaztion in cell types. RNA from whole organs can not give a precise gene expression level: different organs have different ratio of cell types. That's why it is not clear about mRNA level in specific cell types or ratio between cell types (cotyledon and young leaf, for example)-

Please, mention this in discussion.

Detailed comments:

Lines 56 and 70 are very similar, please, remove repetition.

Line 272: „meristem expression“ ?? what do you mean as this?

Line 397: „Quantitative PCR analysis of NGA genes“ maybe you mean gene expression?

Line 402: „secondary leaf“ ??

Figure 7 is quite confusing. Please, provide full description oft he panels in the legends. Moreover, it is not very correct to compare different organs in different secies. Each organ has different cell types with different gene expression profile in cell type. And ratio between cell type in different organs are different.

It will be nice to show parterns of NGA gene localization in situ because only such patterns can led to correct conclusions.

Line 479: „NGA family regulate the YUC genes involved in auxin biosynthesis“ – please, clarify which YUC gene do you mean and in which cell type they are located. How this localization linked with NGA gene expression localization?

Line 483: „Altogether, our results suggest that light might play an essential role in the NGA gene regulation.“ – this sensitence is not logical here.

Lines 589-590: exact repetition.

Moreover, it is well-known that hormonal signaling and, therefore, flower development, leaf morphogenesis were strictly dependent from conditions, including nutrient balance. However, it is very interesting that authors used very low phosphate and „replacament“ it with chloride, which, in turn, may inhibited phospate uptake additionaly. It will be great that authors can explain such decision.

Lines 591-593: can authors explain how they can got flowers in Arabidopsis after 7 days and in tomato after 10 days?? Moreover, 7 days old cotyledon in Arabidopsis and 10 days old tomato have a different biological age and different auxin and other hormonal signaling and can not be directly compared.

Line 603: „Ubiquitin and RNaseH are used as the internal controls for Arabidopsis while actin and ubiquitin are used for tomato“. – please, explain why you choose this gene as housekeeping one.

Author Response

The text is improved, but some questions still need to be answered.

We are grateful to review for careful reading, helpful comments, and constructive suggestions, which has significantly improved our manuscript. We have carefully considered all comments from the reviewers and revised our manuscript accordingly. A point-by-point response to this reviewer’s comments is given below.

The most problematic point is NGA gene localization in cell types. RNA from whole organs cannot give a precise gene expression level: different organs have different ratio of cell types. That's why it is not clear about mRNA level in specific cell types or ratio between cell types (cotyledon and young leaf, for example).

Thank you for the encouraging response. Regarding the gene expression, this is just a preliminary study to examine the NGATHA expression in different organs of Arabidopsis and tomato, as the expression of NGATHA is not yet reported in species other than Arabidopsis.

Furthermore, qPCR was used in the current study to quantify gene expression. The calculation of the relative expression of the target genes by qPCR is based on reference gene(s) as endogenous control(s). The internal controls (reference genes) used in the qRT-PCR are housekeeping genes (endogenous genes), whose expression remains the same throughout the developmental stages, different tissues and different environmental (experimental) conditions. So, ideally, by performing qRT-PCR using a particular housekeeping gene as an internal control the Ct value remains the same in all experimental conditions.

Please, mention this in discussion.

As suggested, we have included this point in the discussion: “However, localization of NGA proteins would provide a better evidence for protein expression in different cell types rather than gene expression studies.”

Lines 56 and 70 are very similar, please, remove repetition.

The sentence is removed.

Line 272: „meristem expression“ ?? What do you mean as this?

The CAT-box was observed in the promoter elements of NGA genes and this cis-element is implicated to be involved in meristem development. Instead of meristem expression, it is now mentioned as meristem development.

Line 397: „Quantitative PCR analysis of NGA genes“ maybe you mean gene expression?

For clarification, we modified the sub-headline as follows: “Expression analysis of NGA genes by qPCR”

Line 402: „secondary leaf“ ??

Secondary leaf is now corrected as “mature leaf”.

Figure 7 is quite confusing. Please, provide full description of the panels in the legends. Moreover, it is not very correct to compare different organs in different species. Each organ has different cell types with different gene expression profile in cell type. And ratio between cell type in different organs are different.

As suggested we modified the legends of Figure 7 for clarification. We agree that each cell type in an organ has a different expression profile, however, in this study, the expression of NGA genes in different organs is examined by qPCR. Furthermore, here, we are not comparing Arabidopsis and tomato as these are two different species.

It will be nice to show patterns of NGA gene localization in situ because only such patterns can led to correct conclusions.

Thank you for your suggestion. It is definitely a good idea to check the NGA gene localization pattern but we are planning to include this in future studies.

Line 479: „NGA family regulate the YUC genes involved in auxin biosynthesis“ – please, clarify which YUC gene do you mean and in which cell type they are located. How this localization linked with NGA gene expression localization?

Modified as follow: “NGA family regulate the AtYUC2 and AtYUC4 genes involved in auxin biosynthesis“.

Line 483: „Altogether, our results suggest that light might play an essential role in the NGA gene regulation.“ – this sentence is not logical here.

The sentence is removed.

Lines 589-590: exact repetition.

Removed as suggested.

Moreover, it is well-known that hormonal signaling and, therefore, flower development, leaf morphogenesis was strictly dependent from conditions, including nutrient balance. However, it is very interesting that authors used very low phosphate and „replacament“ it with chloride, which, in turn, may inhibited phospate uptake additionaly. It will be great that authors can explain such decision.

We used chlorine to sterile the surface of seeds.

Lines 591-593: can authors explain how they can got flowers in Arabidopsis after 7 days and in tomato after 10 days?? Moreover, 7 days old cotyledon in Arabidopsis and 10 days old tomato have a different biological age and different auxin and other hormonal signaling and can not be directly compared.

In the current study, we have not compared Arabidopsis and tomato plants. In Arabidopsis, cotyledons, rosetta leaf, mature leaf and flowers were collected from 7, 21, 28 and 32 days old plants, respectively. In tomatoes, cotyledons, developing young leaves (meristem) and developed mature leaves (second node from the ground) were collected from 12, 28, and 35 days old plants, respectively. The flowers are obtained from 1st set of flowers in both Arabidopsis and tomato. For clarification, we modified the text accordingly.

Line 603: „Ubiquitin and RNaseH are used as the internal controls for Arabidopsis while actin and ubiquitin are used for tomato“. – please, explain why you choose this gene as housekeeping one.

The actin, ubiquitin and RNaseH are the most commonly used housekeeping genes in the molecular labs for gene expression analysis using qPCR. However, the actin primers did not work well with Arabidopsis, as the ct values were not consistent. Therefore, we preferred ubiquitin and RNaseH for Arabidopsis while actin and ubiquitin for tomato.

Quantitative real-time PCR (q-PCR) is one of the most important techniques for gene expression analysis in molecular-based studies. Selecting a proper internal control gene (reference gene) for normalizing data is a crucial step in gene expression analysis. The expression levels of reference genes should remain constant among cells in different tissue. Generally, internal control can be endogenous, which are normal cellular gene sequences whose expression is expected to be constant among different tissue and environmental (experimental) conditions. They are called housekeeping genes. Examples of endogenous internal control genes that have been widely used for PCR process control monitoring include 18s rRNA, ß-actin or GAPDH genes (https://doi.org/10.1038/s41598-019-53544-0).

Round 3

Reviewer 2 Report

Thank you! Plesae, provide some more details.

„Furthermore, qPCR was used in the current study to quantify gene expression. The calculation of the relative expression of the target genes by qPCR is based on reference gene(s) as endogenous control(s). The internal controls (reference genes) used in the qRT-PCR are housekeeping genes (endogenous genes), whose expression remains the same throughout the developmental stages, different tissues and different environmental (experimental) conditions. So, ideally, by performing qRT-PCR using a particular housekeeping gene as an internal control the Ct value remains the same in  all experimental conditions.“

Thanks fort he explanations!

But how can you distinquish between upregulation of gene expression in specific cell (I expect that NGA mRNA level are not the same in mesophyl cell and vasculature, in dividing and differentiated cell): both high level in specific cell type and more cell expressing NGA can give you the same results in qPCR, but these cases have a different meaning, indeed. Please, consider this for the future.

„As suggested we modified the legends of Figure 7 for clarification. We agree that each cell type in an organ has a different expression profile, however, in this study, the expression of NGA genes in different organs is examined by qPCR. Furthermore, here, we are not comparing Arabidopsis and tomato as these are two different species.“

Thanks, It is clear that you did not compare Arabidopsis and tomato, but how reliable comparison cotyledon with polyplid cell with flowers what conatin different cell types? Of course, some information can be extracted, but fort he precise conclusiones more in situ data are required. Maybe 10 days old cotyledon give youa  different resukts, beacse ihave a different develpmental stages Moreover, it is not clear what do you mean as rosette leaf at 21 days. Did you collect inly old one or mix all leaves?

„We used chlorine to sterile the surface of seeds.“-

Thanks, but I mean different. In your growth media you „replace“ phosphate as major anoin invloved in regukation of gene expression with chloride what is toxic (3 mM Cl anion but only 0,625 mM P). This way you induce artifical P deficiency and may significantly chang gene expression profile. Moreover, you use N:P ratio 40:1, what is also toxic for Arabidopsis in particular. Normal N:P must be 5:1.

Please, provide detailed explanation why you preoare such exotic nutrient balance in your experiments.

Author Response

Thank you! Plesae, provide some more details.

But how can you distinquish between upregulation of gene expression in specific cell (I expect that NGA mRNA level are not the same in mesophyl cell and vasculature, in dividing and differentiated cell): both high level in specific cell type and more cell expressing NGA can give you the same results in qPCR, but these cases have a different meaning, indeed. Please, consider this for the future.

Answer: Thank you very much for the valuable suggestion. We will surely look into the NGA gene expression of specific cell types. 

Thanks, It is clear that you did not compare Arabidopsis and tomato, but how reliable comparison cotyledon with polyplid cell with flowers what conatin different cell types? Of course, some information can be extracted, but fort he precise conclusiones more in situ data are required. Maybe 10 days old cotyledon give youa  different resukts, beacse ihave a different develpmental stages Moreover, it is not clear what do you mean as rosette leaf at 21 days. Did you collect inly old one or mix all leaves?

Answer: Thank you for your valuable suggestion. We agree and we will definitely do it! Here we only used relative gene expression. We collected only old leaves from rosetta stage at 21 days.

Thanks, but I mean different. In your growth media you „replace“ phosphate as major anoin invloved in regukation of gene expression with chloride what is toxic (3 mM Cl anion but only 0,625 mM P). This way you induce artifical P deficiency and may significantly chang gene expression profile. Moreover, you use N:P ratio 40:1, what is also toxic for Arabidopsis in particular. Normal N:P must be 5:1.

Please, provide detailed explanation why you preoare such exotic nutrient balance in your experiments.

Answer: Chlorine gas is used for seed surface sterilization (of Arabidopsis  thaliana) as practiced by our lab and many researchers (https://doi.org/10.21769%2FBioProtoc.2685; https://doi.org/10.1111/ppl.13079).  The seeds of Micro-Tom were surface sterilized with 4% (w/v) sodium hypochlorite for 5-10 min and were washed under running water to remove traces of hypochlorite on the seeds. The seeds are spread on wet germination paper and were incubated in dark for germination. Therefore, it is clear that we did not use any chemical in the growth medium.

Our only growth media was a simple Murashige and Skoog medium (or MSO or MS0 (MS-zero)), a plant growth medium that is generally used in the laboratories for the cultivation of plant cell culture.

Major salts (macronutrients) per litre

  • Ammonium nitrate (NH4NO3) 1650 mg/l
  • Calcium chloride (CaCl22H2O) 440 mg/l
  • Magnesium sulfate (MgSO47H2O) 370 mg/l
  • Monopotassium phosphate (KH2PO4) 170 mg/l
  • Potassium nitrate (KNO3) 1900 mg/l.

Minor salts (micronutrients) per litre

  • Boric acid (H3BO3) 6. 2 mg/l
  • Cobalt chloride (CoCl26H2O) 0.025 mg/l
  • Ferrous sulfate (FeSO47H2O) 27.8 mg/l
  • Manganese(II) sulfate (MnSO44H2O) 22.3 mg/l
  • Potassium iodide (KI) 0.83 mg/l
  • Sodium molybdate (Na2MoO42H2O) 0.25 mg/l
  • Zinc sulfate (ZnSO47H2O) 8.6 mg/l
  • Ethylenediaminetetraacetic acid ferric sodium (FeNaEDTA) 36.70 mg/l
  • Copper sulfate (CuSO45H2O) 0.025 mg/l

Vitamins and organic compounds per litre

  • Myo-Inositol 100 mg/l
  • Nicotinic Acid 0.5 mg/l
  • Pyridoxine HCl 0.5 mg/l
  • Thiamine HCl 0.1 mg/l
  • Glycine 2 mg/l
  • Tryptone 1 g/l (optional)
  • Indole Acetic Acid 1-30 mg/l (optional)
  • Kinetin 0.04-10 mg/l (optional)

Round 4

Reviewer 2 Report

Thank you for the explanation.

But: "Therefore, it is clear that we did not use any chemical in the growth medium."?

Macro and micro salt is a chemical.

Moreover, you only copy-paste composition of the medium for callus, but did not provide any explanation why you choose it based on function of each nutrients!

Your medium have 6 mM Cl (what  does not have any major function in plants) and very toxic in the case of Arabidopsis: it rapidly uotake ba vaciole and led to abnormal cell expansion.

In the same time you reduce phosphate (major anion with many vital functions) to only 1,25 mM and use N:P ratio 60:1,25 instead 5:1 (what is optimal for Arabidopsis).

Overally, it may mean that you study gene expression under high nutritional stress.

pleasem provide brief explanation.

The rest part is excellent.

Author Response

Thank you very much for your attention. We used half-strength Murashige and Skoog medium (MS) with 1% sucrose, which means using half the quantity of macro and micronutrients. Since, In the current study, seedlings were grown in the same half-strength MS medium, the conditions for all seedlings are similar.

Round 5

Reviewer 2 Report

Thank you very much for discussion and answer. It is very inetersting to learn that nutrional stress does not has effect on NGATHA gene expression in your case. Please, be very carefully next time when choosing growth conditions because stress (including N:P 40:1) have a very different effects on gene expression in wt and mutants.

Author Response

Dear reviewer,

Thank you very much for the careful review. We will definitely take care of the growth media and conditions used for the study.

Thank you,

Dr. Hymavathi